# Nutritional and laboratory parameters affect the survival of dogs with chronic kidney disease

Vivian Pedrinelli[1], Daniel Magalhães Lima[2], Caio Nogueira Duarte[1], Fabio Alves Teixeira[1], Mariana Porsani[1], Cecilia Zarif[1], Andressa Rodrigues Amaral[1], Thiago Henrique Annibale Vendramini[3], Marcia Mery Kogika[4], Márcio Antonio Brunetto[1,3]*

1 Veterinary Nutrology Service, Teaching Veterinary Hospital, School of Veterinary Medicine and Animal Science, University of Sao Paulo (USP), Sao Paulo, Brazil, 2 Department of Preventive Veterinary Medicine and Animal Health, School of Veterinary Medicine and Animal Science, University of Sao Paulo (USP), Sao Paulo, Brazil, 3 Animal Nutrition and Production Department, Pet Nutrology Research Center, School of Veterinary Medicine and Animal Science, University of Sao Paulo (USP), São Paulo, Pirassununga, Brazil, 4 Department of Internal Medicine, Small Animal Internal Medicine Service, School of Veterinary Medicine and Animal Science, University of Sao Paulo (USP), Sao Paulo, Brazil

* mabrunetto@usp.br

**Data Availability Statement:** All relevant data are within the manuscript.

**Funding:** The author(s) received no specific funding for this work.

## Abstract

Chronic kidney disease is a common disease in dogs, and factors such as serum concentrations of creatinine, albumin, and phosphorus at the moment of diagnosis may influence the survival of these patients. The present retrospective study aimed to evaluate the relationship between survival in dogs with chronic kidney disease and laboratory parameters (creatinine, phosphorus, albumin, and hematocrit) and nutritional parameters (body condition score, muscle mass score, type of food, appetite and feeding method). A total of 116 dogs with chronic kidney disease stages 2 to 4 were included, and survival was calculated considering the time between diagnosis and death. Survival curves were configured by Kaplan-Meier analysis and a comparison between survival curves was performed by the log-rank test. Factors related to survival were disease stage (p<0.0001), serum phosphorus concentration (p = 0.0005), hematocrit (0.0001), body condition score (p = 0.0391), muscle mass score (p = 0.0002), type of food (p = 0.0009), feeding method (p<0.0001) and appetite (p = 0.0007). Based on data obtained in this study, it is possible to conclude that early diagnosis, as well as nutritional evaluation and renal diet intake, are determinant strategies to increase survival in dogs with chronic kidney disease.

## Introduction

Chronic kidney disease (CKD) is considered one of the most common diseases in dogs, and its prevalence can vary between 0.5 to 3.0% of the general population and can be up to 10.0% on the hospitalized canine population [1,2]. It is a progressive disease caused by morphological and functional changes of the kidneys, and clinical signs usually occur when there is more

**Competing interests:** The authors have declared that no competing interests exist.

than 70.0% of nephrons compromised, and it can be congenital or acquired, the latter being more common in animals of more than 7 years of age [3–7].

The diagnosis is based on the animal's history, clinical signs and laboratory exams. The main clinical signs in dogs with CKD are polyuria, polydipsia, emesis, and muscle weakness, which are the result of the decrease in the glomerular filtration capacity and the accumulation of toxic substances [6,7]. The most common laboratory changes in this disease are azotemia, hyperphosphatemia, metabolic acidosis, non-regenerative anemia, decreased urinary specific gravity (isosthenuria) and proteinuria [8,9]. Changes in imaging exams, such as ultrasound, are also an important diagnostic tool, especially for the characterization of congenital alterations [6,7]. To focus on the needs of different CKD stages, the International Renal Interest Society [10] suggests four stages.

Several factors can influence the progression rate of CKD and, therefore, can influence survival. One of these factors is the increase of serum phosphorus, which leads to calcium phosphate deposit in soft tissues, including renal cells, and thus can increase the loss of functional cells [3]. Another factor that can influence progression rate is the normocytic normochromic non-regenerative anemia, which occurs mainly because of decreased synthesis of erythropoietin by the kidneys, and can lead to hyporexia or anorexia and reduce general health and survival [6,11].

Another change common in CKD is hypoalbuminemia, and the main cause in this disease is glomerular protein loss inflammation, and chronic malnutrition [6, 12]. Previous studies have correlated serum albumin concentration with survival in dogs. Michel [13] observed that hospitalized animals with albumin concentrations below 2.7g/dL presented lower hospital discharge rates than animals with concentrations above this value. Regarding survival in dogs with CKD, two studies observed decreased survival in patients with hypoalbuminemia when compared to those with values between the reference ranges of each study [14,15].

Associated with conservative treatment, nutrition is considered an important tool in the management of CKD and, therefore, there is a recommendation to introduce a therapeutic renal diet as of stage 2 [6,8,16]. Nutritional key aspects for patients with CKD are to help control clinical signs of the disease, to reduce electrolytic and mineral disturbances, and to maintain muscle mass score (MMS) and body condition score (BCS) [7]. However, it must be taken into consideration that anorexia and hyporexia are common consequences of CKD and they occur because of the series of metabolic alterations caused by the disease [7,8]. For dogs that, even with support treatment, do not voluntarily ingest sufficient food quantities to supply their energy and nutritional requirements, feeding tubes should be recommended [1,17,18].

There is scarce evidence about factors that influence survival in dogs with CKD. In a retrospective study conducted by Parker and Freeman [14], 100 dogs with CKD were evaluated and it was observed that dogs diagnosed in stage 2 had increased survival than those diagnosed in stages 3 and 4. Furthermore, it was observed that underweight dogs presented decreased survival than those of ideal weight or those who were overweight. A recent prospective study conducted by Rudinsky et al. [15] evaluated the influence of factors on the survival of 27 dogs with CKD. Factors such as hypoalbuminemia, hyperphosphatemia, low BCS and muscle mass loss were associated with decreased survival in these animals.

As for the influence of the type of diet on the survival of dogs with CKD, one study assessed two groups for 24 months, one that received a therapeutic renal diet and one that received a maintenance diet [19]. During the period of the study, dogs that received the renal diet took 2.5 times more time to develop uremic crisis, and survival of this group was almost three times higher than the group that received a maintenance diet.

Based on the current scarce evidence regarding parameters like type of diet, BCS, MMS, and serum concentrations of substances linked to CKD and the survival of dogs with this

disease, the present study aimed to evaluate the influence of laboratory and nutritional parameters in the survival of dogs with CKD.

## Materials and methods

This study was approved by the Animal Use Ethics Committee from the School of Veterinary and Animal Science of the University of Sao Paulo (FMVZ/USP), protocol number 3138/2013. Information was obtained retrospectively from records of dogs assessed between February 2013 and December 2018 by the Veterinary Nutrology Service of the Teaching Hospital of the School of Veterinary Medicine and Animal Science from the University of Sao Paulo, Brazil. Inclusion criteria were dogs with 12 months of age or older, with serum creatinine concentrations above 1.4mg/dL for more than three months and urine specific gravity of 1.030 or lower, which characterizes animals diagnosed with CKD stages 2 and up [10]. Exclusion criteria were dogs with comorbidities at the moment of diagnosis and animals with incomplete records or with records that did not contain the information necessary to this study, and dogs that were already treated before the assessment. Furthermore, gestating and lactating bitches and dogs diagnosed with congenital kidney diseases were not included.

Information obtained from the records at the time of diagnosis were: age; breed; body weight; BCS [20]; MMS [21]; hyporexia or anorexia; if feeding tubes were used; serum concentrations of creatinine, albumin, phosphorus, and urea; and percentage of hematocrit.

Animals were considered in hyporexia if the daily intake of food was reduced in 25% or more, and were considered in anorexia if the owner reported no food intake and the animal did not accept food (dry and/or wet) offered by the staff of the Nutrition Service at the first assessment.

Data regarding the type of food consumed from the time of diagnosis until death was also collected.

For calculation of survival time, the date of death was obtained from the hospital records or, if not presented in the records, after a telephone contact with the owners. When the date of death was impossible to obtain or if the animal was still alive after the end of the data collection period, this information was considered as censored data and the date considered for the survival curve was from the last assessment performed in the hospital.

Statistical analysis was performed with GraphPad Prism 6.0 (GraphPad Software, USA). Survival curves were calculated with Kaplan-Meier analysis and log-rank test (Mantel-Cox) was used to compare curves. Values of $p \leq 0.05$ were considered significant.

## Results

A total of 271 dog records were assessed, and 116 animals were included according to the study's criteria, 59 females and 57 males, with a mean age of 11.4±3.5 years (range 3 to 20 years) and a mean body weight of 13.9±11.4kg (range 1,8 to 54,1 kg) at the moment of diagnosis. The 155 animals that were not included either had comorbidities at the time of diagnosis (n = 134; 86.45%) or did not have complete record information (n = 21; 13.55%). Breeds included mixed breed (n = 40), Poodle (n = 17), Labrador Retriever (n = 9), English Cocker Spaniel (n = 7), Lhasa Apso (n = 6), Miniature Schnauzer (n = 6), Pinscher (n = 5), Dachshund (n = 4), Golden Retriever (n = 4), Yorkshire Terrier (n = 4), Shih Tzu (n = 3), Weimaraner (n = 2) and one of each of the following: Beagle, Boxer, Brazilian Terrier, Bull Terrier, German Shepherd, Kuvasz, Maltese, Pug, and White Swiss Shepherd.

Six dogs were still alive at the time of the data analysis and 16 dogs did not have an established date of death. Of the 94 dogs with death dates, 83 were euthanized due to complications

**Table 1. Distribution of dogs with chronic kidney disease according to laboratory and nutrition parameters and their influence on survival.**

| Item | | N. dogs (%) | Survival median (days) | p |
|---|---|---|---|---|
| Stage[1] | 2 | 50 (43.1) | 475[a] | <0.0001 |
| | 3 | 43 (37.1) | 187[b] | |
| | 4 | 23 (19.8) | 13[c] | |
| BCS[2] | 1 to 3 | 48 (41.4) | 125[a] | 0.0391 |
| | 4 and 5 | 35 (30.2) | 103[a] | |
| | 6 to 9 | 33 (28.4) | 327[b] | |
| MMS[3] | 0 | 9 (7.8) | 45[a] | 0.0002 |
| | 1 | 34 (29.2) | 122[a] | |
| | 2 | 51 (44.0) | 206[b] | |
| | 3 | 22 (19.0) | 255[c] | |
| Serum phosphorus[4] | ≤ 5.5mg/dL | 31 (48.4) | 573 | 0.0005 |
| | > 5.5mg/dL | 33 (51.6) | 136 | |
| Hematocrit[4] | ≤ 37% | 59 (53.6) | 99 | 0.0001 |
| | > 37% | 51 (46.4) | 415 | |
| Serum albumin[4] | ≤ 2.3g/dL | 18 (17.5) | 128 | 0.2260 |
| | > 2.3g/dL | 85 (82.5) | 263 | |
| Appetite | Anorexia | 38 (32.8) | 33[a] | 0.0007 |
| | Hyporexia | 38 (32.8) | 246[b] | |
| | No alteration | 40 (34.4) | 429[b] | |
| Feeding method | Feeding tube | 41 (35.4) | 32 | <0.0001 |
| | Voluntary | 75 (64.6) | 309 | |
| Type of food[5] | Renal[6] | 63 (54.3) | 309 | 0.0009 |
| | Other | 53 (45.7) | 92 | |

[1]Staging according to IRIS [10]

[2]Body condition score according to Laflamme [20]

[3]Muscle mass score according to Michel et al. [21]

[4]Data not present for all dogs in the study

[5]Type of food consumed for 75% of the survival time or more

[6]Commercial or homemade diet formulated by the veterinarians of the Nutrology Service team of the hospital (16% crude protein and 0.3% phosphorus in dry matter basis)

[a,b,c]Medians with more than two variables with different letters were statistically different (p≤0.05).

of the disease and 11 died at their household. The distribution of animals according to parameters is described in Table 1.

Stage of the disease, serum phosphorus concentration, BCS, MMS, and hematocrit at the moment of diagnosis influenced survival (Figs 1–5). Serum albumin concentration at diagnosis, however, did not influence survival time (Fig 6).

The feeding method and appetite at diagnosis also influenced survival, although there was no difference between dogs with hyporexia and no appetite alteration (p = 0.1080) but a difference between dogs with anorexia and all the others (p = 0.0088) (Figs 7 and 8).

Type of food prescribed at the time of diagnosis were therapeutic renal diets (n = 28; 24.1%), homemade diets with phosphorus and protein restriction (n = 29; 25.0%), commercial senior diets (n = 12; 10.4%), homemade diet for maintenance (n = 9; 7.8%), and other commercial diets (n = 15; 12.9%). Furthermore, milk replacers (n = 6; 5.2%) and a commercial powdered enteral feeding diet (n = 17; 14.6%) were prescribed for animals that had feeding tubes placed at the time of diagnosis. As for the type of diet consumed for 75% or more of the

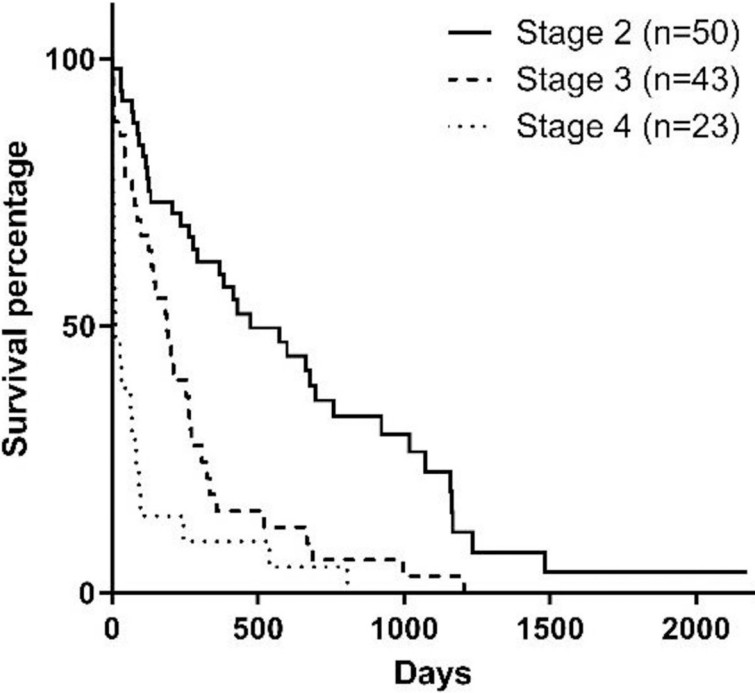

**Fig 1. Survival curve according to CKD stage [10] at diagnosis (p<0.0001).**

time between diagnosis and death, animals that consumed renal diets (commercial or home-made) presented increased survival time when compared to those who consumed other diets (Fig 9).

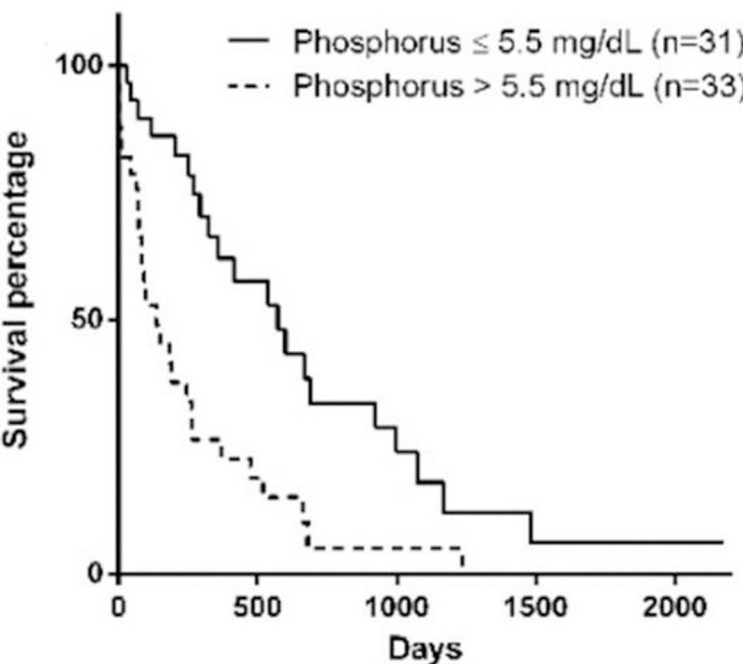

**Fig 2. Survival curve according to serum phosphorus at diagnosis (p = 0.0005).**

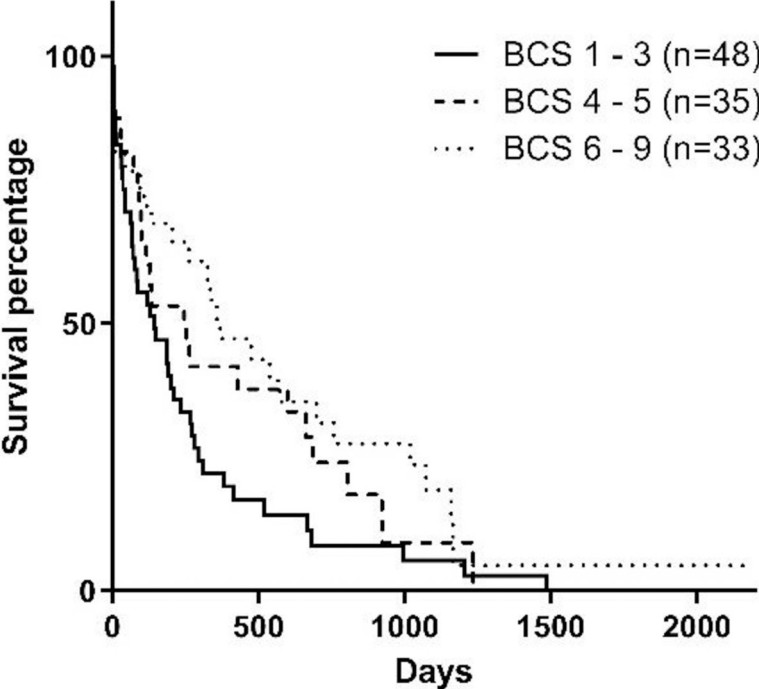

**Fig 3. Survival curve according to BCS [20] at diagnosis (p = 0.0391).**

The distribution of dogs by appetite and feeding method at diagnosis according to CKD stage is presented in Table 2.

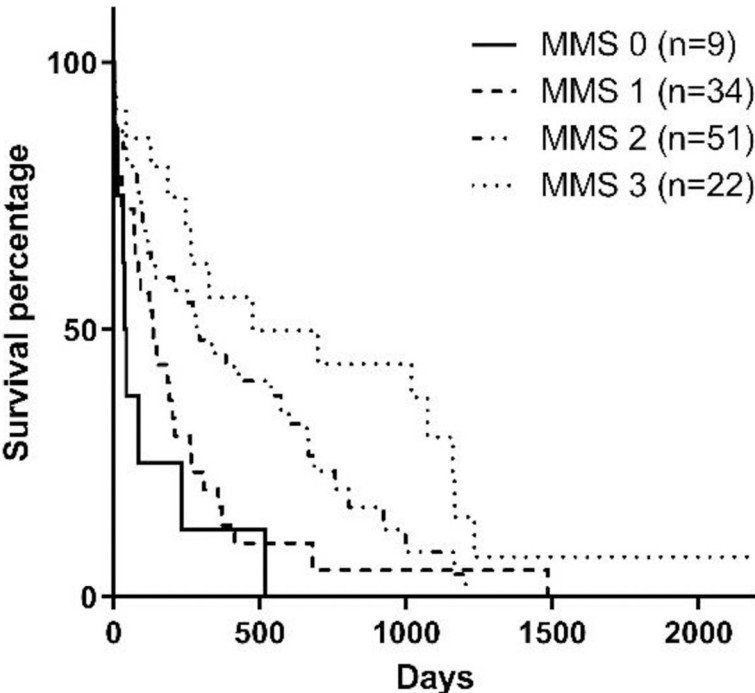

**Fig 4. Survival curve according to MMS [21] at diagnosis (p = 0.0002).**

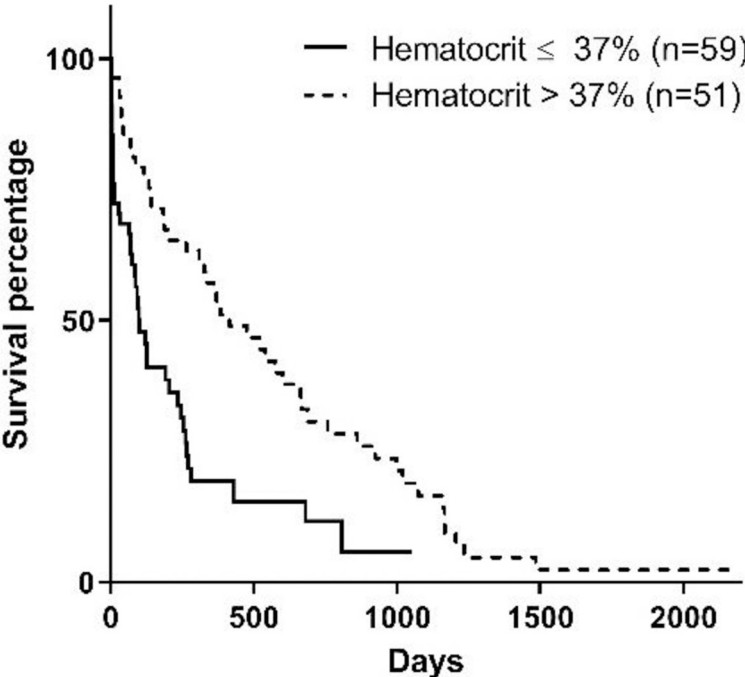

**Fig 5. Survival curve according to the hematocrit at diagnosis (p = 0.0001).**

## Discussion

In the present study, the factors that were associated with increased survival in dogs with CKD were early stages of the disease, higher BCS, higher MMS, phosphorus levels and hematocrit in

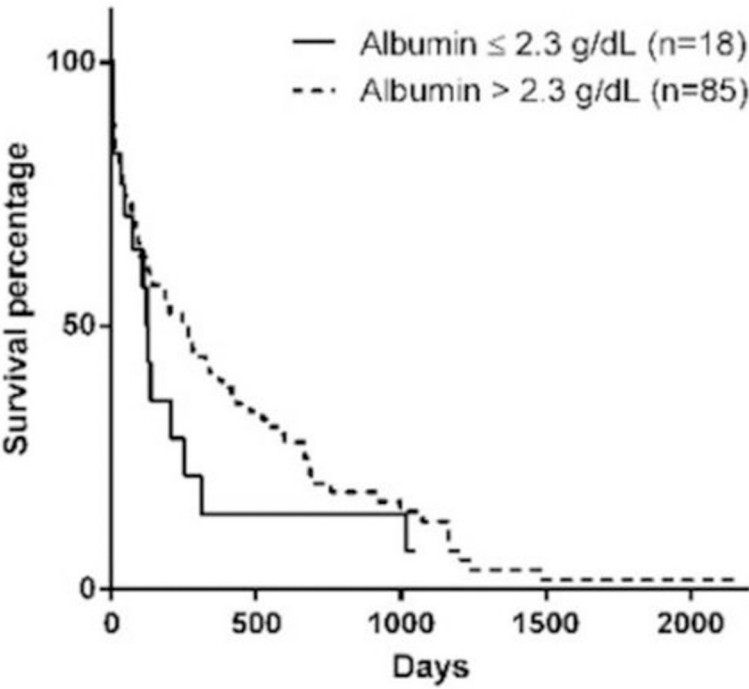

**Fig 6. Survival curve according to serum albumin at diagnosis (p = 0.2260).**

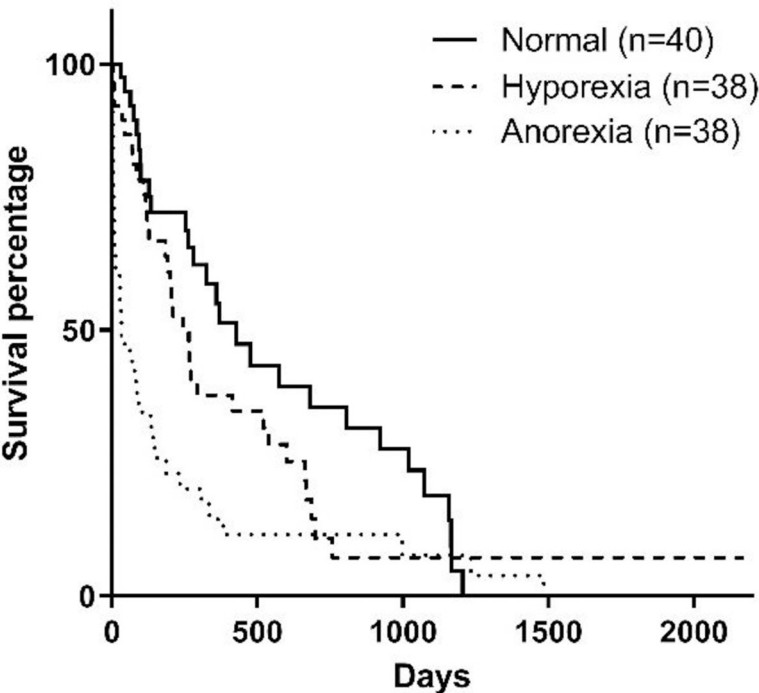

**Fig 7. Survival curve according to appetite at diagnosis (p = 0.0007).**

the reference ranges, along with the intake of renal diets, no alteration in appetite or hyporexia and voluntary feeding.

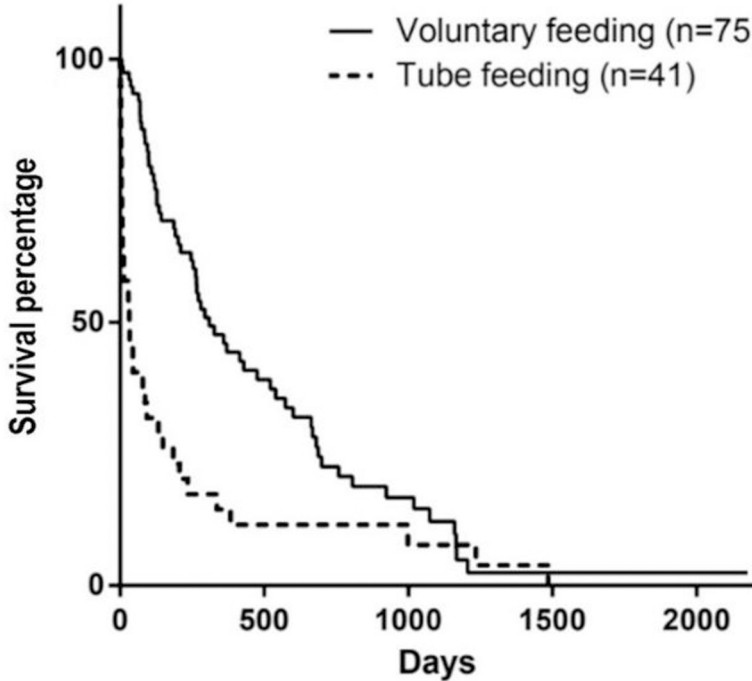

**Fig 8. Survival curve according to the feeding method at diagnosis (p<0.0001).**

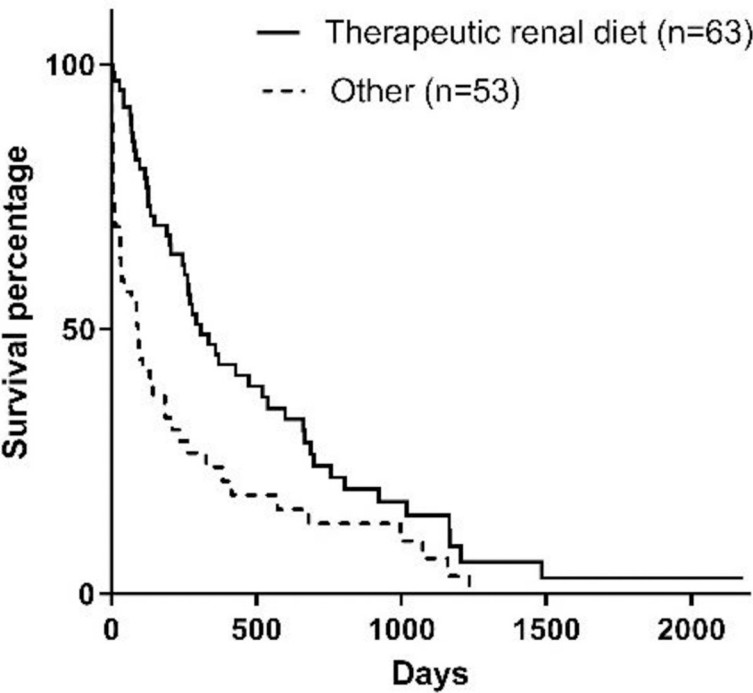

**Fig 9. Survival curve according to the type of food of 75% or more of the period from diagnosis until death (p = 0.0009).**

The stage of the disease had a negative correlation with survival in the studied population, which is similar to previous studies [14,15]. This is an important fact because it demonstrates that early diagnosis of CKD increases considerably the survival of the animal. This may be due to early conservative treatment, which can help correct dehydration, metabolic acidosis, and hyperphosphatemia, as well as the change to a renal diet, which was also associated with increased survival.

The body weight was not evaluated in the present study, but instead, BCS and MMS were evaluated, considering that even if the body weight changes, it can underestimate cachexia or include ascites or edema. Therefore, body weight is not considered to be specific, as observed by Parker and Freeman [14]. Overweight and obese dogs (BCS 6 to 9) presented increased survival than dogs with ideal body weight (BCS 4 and 5) or underweight dogs (BCS 1 to 3) [20], similar to studies conducted by Parker and Freeman [14] and Rudinsky et al. [15]. This reverse relationship between body condition and survival time is known as the obesity paradox, and is been previously observed in dogs with heart failure [22] and in humans, both with heart failure

**Table 2. Distribution of dogs by appetite and feeding method according to the stage of chronic kidney disease at the time of diagnosis.**

| Item | | | Appetite | | | Feeding method | |
|---|---|---|---|---|---|---|---|
| | | | Anorexia (%) | Hyporexia (%) | No alteration (%) | Voluntary (%) | Feeding tube (%) |
| Stage[1] | | 2 | 9 (18.0) | 17 (34.0) | 24 (48.0) | 41 (82.0) | 9 (18.0) |
| | | 3 | 15 (34.9) | 15 (34.9) | 13 (30.2) | 28 (65.1) | 15 (34.9) |
| | | 4 | 14 (60.9) | 6 (26.1) | 3 (13.0) | 9 (39.1) | 14 (60.9) |

[1]Staging according to IRIS [10].

and advanced chronic kidney disease [23,24]. Among the potential reasons for this paradox in patients with heart conditions is the effect of endocrine factors secreted by the adipose tissue [24]. However, the cause of this paradox in dogs with CKD is yet to be determined.

Several chronic diseases can lead to weight loss and also muscle mass loss, including CKD [18]. The BCS is used to evaluate body fat deposits and, therefore, it must be used together with a tool to evaluate muscle mass, once there is no clinical correlation between these two evaluations [20,21]. In the present study, dogs at the time of diagnosis of CKD that presented mild, moderate or severe muscle loss had lower survival when compared to dogs that did not present muscle loss. Another study evaluated the association between MMS and survival in dogs with CKD, with similar results as the present study [15]. This study, however, divided animals into only two groups according to muscle mass: animals with muscle loss and animals without muscle loss. The present study presents an analysis according to the MMS published by Michel et al. [21], which enables a more precise evaluation of the correlation between MMS and survival in dogs with CKD.

Hypoalbuminemia was not correlated to survival in the present study, which differs from previous studies conducted on dogs with CKD [14,15]. The difference observed between the studies may be attributed to the laboratory reference ranges used. In the present study, the minimum reference value for albumin was considered as 2.3g/dL, which is the reference value used in the teaching hospital where the study took place. Rudinsky et al. [15] used a minimum reference value of 2.9g/dL, and in the study conducted by Parker and Freeman [14], this information was not available.

Another parameter that had a negative correlation with survival was the serum phosphorus concentration, similar to the results observed by Rudinsky et al. [15]. The post-treatment serum phosphorus concentrations for dogs with CKD recommended by IRIS [16] are 4.6mg/dL, 5.0mg/dL, and 6.0mg/dL for stages 2, 3 and 4, respectively. To achieve these concentrations, diets with controlled or restricted phosphorus are recommended, as well as the use of phosphate binders [8,16].

Regarding hematocrit percentage, dogs that had values under 37.0% in the present study had 4 times lower survival time than animals with hematocrit values of 37.0% or higher. This differs from results observed by Rudinsky et al. [15], and may be due to two reasons: first, the previous study considered 36.0% of hematocrit; and second, the number of animals in the previous study was 27, as opposed to 116 dogs included in the present study.

As for the influence of diet, Jacob et al. [19] conducted a prospective study for 24 months and observed that dogs with CKD that consumed a therapeutic renal diet had survival time 3 times higher than those that consumed maintenance diets. The results of the present study corroborate these findings. It is important to state that some of the animals in the present study were prescribed diets that were not therapeutic. This occurred when dogs did not accept the renal diet or owners could not afford to purchase the therapeutic diets. However, as seen in the present study and in the study conducted by Jacob et al. [19], the renal diet is the best option for dogs with CKD in terms of survival.

Another factor related to nutrition observed in the present study is that dogs that were anorexic at the moment of diagnosis had lower survival time than dogs with hyporexia or with no changes in appetite. This can be justified considering that anorexia is more common in dogs with advanced stages of CKD, as observed in the present study, which has also been correlated to lower survival [1,8,15]. The feeding method prescribed at the moment of diagnosis did also influence survival. Animals that had feeding tubes placed were mostly in stages 3 and 4, and similar to appetite, the need to use feeding tubes is associated with disease stage [18]. Therefore, the influence of the appetite of the dog at the time of diagnosis, and consequently

the need for feeding tubes, are important parameters that can influence survival in dogs with CKD.

## Conclusions

Based on the results of the present study it is possible to conclude that early diagnosis is important, so the proper treatment and clinical assessment can be established and, therefore, extend survival. Factors such as BCS and MMS at diagnosis influenced survival, which makes the assessment of these parameters of uttermost importance in dogs with CKD. Furthermore, the intake of a renal diet was determinant to increase survival and should be part of the recommendations for patients with this diagnosis. The present study brings to light important information to better understand the prognosis of CKD and also identify key nutritional points to provide a better quality of life and increase survival in dogs with chronic kidney disease.

## Author Contributions

**Data curation:** Márcio Antonio Brunetto.

**Formal analysis:** Vivian Pedrinelli, Caio Nogueira Duarte, Mariana Porsani, Cecilia Zarif, Andressa Rodrigues Amaral, Thiago Henrique Annibale Vendramini, Marcia Mery Kogika, Márcio Antonio Brunetto.

**Investigation:** Vivian Pedrinelli, Daniel Magalhães Lima, Caio Nogueira Duarte, Fabio Alves Teixeira, Mariana Porsani, Cecilia Zarif, Andressa Rodrigues Amaral, Thiago Henrique Annibale Vendramini, Marcia Mery Kogika, Márcio Antonio Brunetto.

**Methodology:** Caio Nogueira Duarte, Fabio Alves Teixeira, Mariana Porsani, Cecilia Zarif, Andressa Rodrigues Amaral, Thiago Henrique Annibale Vendramini, Marcia Mery Kogika, Márcio Antonio Brunetto.

**Software:** Daniel Magalhães Lima.

**Supervision:** Márcio Antonio Brunetto.

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
