## [Decision Letter · Decision Letter 0]

30 Apr 2020

PONE-D-20-08705

Nutritional and laboratory parameters affect the survival of dogs with chronic kidney disease

PLOS ONE

Dear Dr Brunetto

Thank you for submitting your manuscript to PLOS ONE. After careful consideration, we feel that it has merit but does not fully meet PLOS ONE’s publication criteria as it currently stands. Therefore, we invite you to submit a revised version of the manuscript that addresses the points raised during the review process.

Many thanks for submitting your manuscript to PLOS One

Your manuscript was reviewed by two experts in the field, and they have recommended some minor revisions

If you could write a response to reviewers comments that will help expedite review when it is resubmitted. Please dont feel that you need to respond to each minor grammatical modification, just a comment saying they were done will suffice.

Wishing you the best of luck with your revisions

Hope you are keeping safe and well in these difficult times

Thanks

Simon

We would appreciate receiving your revised manuscript by Jun 14 2020 11:59PM. To enhance the reproducibility of your results, we recommend that if applicable you deposit your laboratory protocols in protocols.io, where a protocol can be assigned its own identifier (DOI) such that it can be cited independently in the future. For instructions see: http://journals.plos.org/plosone/s/submission-guidelines#loc-laboratory-protocols

We look forward to receiving your revised manuscript.

Kind regards,

Simon Russell Clegg, PhD

Academic Editor

PLOS ONE

2. In your Methods section, please provide additional details regarding participant consent from the owners of the animals. In the ethics statement in the Methods and online submission information, please ensure that you have specified (1) whether consent was informed and (2) what type you obtained (for instance, written or verbal). If the need for consent was waived by the ethics committee, please include this information

Reviewers' comments:

Reviewer's Responses to Questions

**Comments to the Author**

1. Is the manuscript technically sound, and do the data support the conclusions?

Reviewer #1: Yes

Reviewer #2: Yes

2. Has the statistical analysis been performed appropriately and rigorously? 

Reviewer #1: Yes

Reviewer #2: Yes

3. Have the authors made all data underlying the findings in their manuscript fully available?

Reviewer #1: No

Reviewer #2: Yes

4. Is the manuscript presented in an intelligible fashion and written in standard English?

Reviewer #1: Yes

Reviewer #2: Yes

5. Review Comments to the Author

Reviewer #1: This is a well-written succinct manuscript that will add nicely to the canine CKD literature. I have just a few comments/questions for the authors to address.

1. Since the cases recruited were from 2013-2018, it seems as though they should be staged according to the IRIS guidelines from 2017 and earlier, as the cut-offs between stages 2 and 3 changed in the 2019 version. Also, I am not convinced that Table 1 is necessary. Most readers should be familiar with IRIS staging, and if not, it is referenced accordingly. I will defer to the Editor on these comments.

2. Did you have IRIS substage information for these dogs? UPC? Blood pressure measurement? Please add this if so or comment otherwise if not. Especially since proteinuria and hypertension have also been shown to correlate with survival in CKD patients.

3. It might be nice to include additional data on the dogs included in another table, including medians and ranges or means and standard deviations. I defer to the Editor on this.

4. Since this was a retrospective study, please comment further on how MMS was determined. Is it standard for this to be recorded in all records? This is unfortunately not true for many academic institutions. Were specific scores provided or did the authors extrapolate MMS scores from the records?

5. Please comment on the use of a numerical feline MMS score (Michel) vs. a canine muscle condition score (eg, AAHA & WSAVA recommendations for using descriptive terms (eg, normal, mild, moderate, severe atrophy). It would be ideal to use a descriptive score here.

6. Please comment further on how dogs were determined to be completely anorexic vs. hyporexic. Was their intake compared to RER or MER? How subjective was it? How many days did that history entail? If this is too muddy, I might suggest combining anorexia and hyporexia into one category for statistical analysis.

7. For dogs that had feeding tubes, how long were they in place?

8. Would you care to comment further on why dogs with higher BCS and MMS lived longer than the underweight and muscle wasted dogs? This relates to the obesity paradox and there is a great deal of literature in people with more coming out in dogs and cats with various diseases.

9. Were there any other differences in the dogs regarding medication and/or dietary supplement use? Please comment further on these results.

10. It is important to recognize that not all “senior” diets are appropriate for dogs with CKD, and their nutrient profiles can vary tremendously. It might be nice to make a statement regarding this so that it is not inferred otherwise.

Specific comments:

1. Lines 69-70. In the context of CKD, it seems prudent to list inflammation and malnutrition as additional factors that could influence hypoalbuminemia.

2. Line 80. Please abbreviate body condition score (BCS) here and then use this abbreviation subsequently (eg, line 116)

3. Line 100. Please denote that MMS refers to muscle mass score here before its first abbreviation. See line 116.

Reviewer #2: You have written a very nice paper here with some very interesting results. I thoroughly enjoyed reading it. One thing which does spring to mind which maybe a study worth undertaking is to define reference ranges for these biochemical parameters which will allow for comparison of studies between countries. I have only made minor points as I think the manuscript is good, and well written, so I both commend you, and thank you.

Line 34- influenced survival?

Line 46- perhaps in the hospitalised dog population?

Line 49- replace ‘with’ with ‘of’

Table 1- is this the most up to date details? I believe that this has been updated (but could be wrong)

I am also not convinced that you need this, but If you wish you can leave it in

Line 75- Maybe state what you class as the reference range as in my experience this seems to vary slightly between countries and even between clinics

Line 89- those of ideal weight

Line 99- you perhaps need to define BCS earlier?

Line 100- possibly also need to define MMS?

Line 110- did you have all the clinical stages of disease for each animal in the study?

Line 116- you have the above two terms defined here- these would be better prior to this

Line 117- maybe better reworded as ‘if feeding tubes were used’ ?

Line 127- p = <0.05

Line 130- looks like a space missing between ‘and 116’

Line 132- I wonder if the mean for the body weight is the best, or would a range be better? As a miniature schnauzer would be much less than a golden retriever? Were any differences seen in biochemical parameters between breeds? Or any difference in survival?

Line 140- at the time of data analysis?

Table 2- did you take into account variation among people for BSC and MMS? That may be very different between animals

Also, how did you define hyporexic and anorexic?

Line 171- also influenced survival?

Line 180- at the time of diagnosis?

Line 180-181- Would it be possible to have more details on the contents of these foods? Perhaps as a supplementary table? If not, I think somewhere it would be nice to define what you mean by a renal diet and what that will include.

I do wonder if the figures would be better as a large block rather than individual? But am happy either way so will leave that up to you

Line 203- no alteration was seen in appetite and …

Line 221- at the time of diagnosis?

Line 229- 235- I think this is an important study which needs to be done at some point, to define exact parameters for this disease which allows for cross study comparison, and similar treatment and diagnostics of dogs between surgeries

Line 243- haematocrit values of 37.0% of higher….

Line 258- I think this section ends very abruptly. Perhaps a nice summary line may be useful?

6. PLOS authors have the option to publish the peer review history of their article (what does this mean?). If published, this will include your full peer review and any attached files.

Reviewer #1: No

Reviewer #2: No

---

## [Author Response · Author response to Decision Letter 0]

26 May 2020

Dear Editor, 

The authors wish to thank you for the attention given to our research. We have considered all the suggestions and comments by the reviewers, which are addressed below. Thank you once again for considering our work for publishing.

Reviewer #1: This is a well-written succinct manuscript that will add nicely to the canine CKD literature. I have just a few comments/questions for the authors to address.

1. Since the cases recruited were from 2013-2018, it seems as though they should be staged according to the IRIS guidelines from 2017 and earlier, as the cut-offs between stages 2 and 3 changed in the 2019 version. Also, I am not convinced that Table 1 is necessary. Most readers should be familiar with IRIS staging, and if not, it is referenced accordingly. I will defer to the Editor on these comments.

Thank you for your comment. We changed the version of IRIS to that of 2017, and removed Table 1.

2. Did you have IRIS substage information for these dogs? UPC? Blood pressure measurement? Please add this if so or comment otherwise if not. Especially since proteinuria and hypertension have also been shown to correlate with survival in CKD patients.

The information regarding UPC and blood pressure was not obtained because most records were not clear on this information, this information in most cases is in the referral’s records. Unfortunately, as this is a retrospective study, we are only able to use what is on our records.

3. It might be nice to include additional data on the dogs included in another table, including medians and ranges or means and standard deviations. I defer to the Editor on this.

Thank you for your comment. We opted to include ranges for age and body weight in the manuscript (lines 140-141), and not as a table.

4. Since this was a retrospective study, please comment further on how MMS was determined. Is it standard for this to be recorded in all records? This is unfortunately not true for many academic institutions. Were specific scores provided or did the authors extrapolate MMS scores from the records?

The MMS is standard for all the patients assessed by the Nutrology Service at our teaching hospital, and that is why we only included animals that were assessed by the nutrition staff. Specific scores were provided according to the scale between 0 and 3 published by Michel et al. (2011). 

5. Please comment on the use of a numerical feline MMS score (Michel) vs. a canine muscle condition score (eg, AAHA & WSAVA recommendations for using descriptive terms (eg, normal, mild, moderate, severe atrophy). It would be ideal to use a descriptive score here.

We determined that the numerical score described by Michel et al. (2011) would be standard for our practice. We believe it does not impact the results of the present study, since the WSAVA recommendations of muscle condition score are based on the research of Michel and coworkers, which was not published then, as cited in the WSAVA Global Nutrition Guidelines (Freeman et al., 2011).

6. Please comment further on how dogs were determined to be completely anorexic vs. hyporexic. Was their intake compared to RER or MER? How subjective was it? How many days did that history entail? If this is too muddy, I might suggest combining anorexia and hyporexia into one category for statistical analysis.

Dogs were determined to be anorexic if the owner reported that the animal did not eat anything at least for a day. When this occurs, it is standard in our practice to offer the animal some types of food (dry and/or wet) to evaluate appetite. If the animal does not accept any food, it is considered anorexic. The dogs were considered hyporexic if the daily intake of food was reduced in 25% or more. We added this to the text to make it more clear to the readers (lines 12-126).

7. For dogs that had feeding tubes, how long were they in place?

It varied greatly between individuals, but it ranged from 1 to 12 days.

8. Would you care to comment further on why dogs with higher BCS and MMS lived longer than the underweight and muscle wasted dogs? This relates to the obesity paradox and there is a great deal of literature in people with more coming out in dogs and cats with various diseases.

Thank you for your comment. We added a paragraph in the discussion section regarding the obesity paradox (lines 229-234).

9. Were there any other differences in the dogs regarding medication and/or dietary supplement use? Please comment further on these results.

There were many differences regarding treatment, especially regarding correction of dehydration and phosphorus chelation drugs. However, as the data compared is that obtained at the time of diagnosis, the treatment did not interfere in the results. As this was not made clear, we added a sentence in the material and methods section (line 115).

10. It is important to recognize that not all “senior” diets are appropriate for dogs with CKD, and their nutrient profiles can vary tremendously. It might be nice to make a statement regarding this so that it is not inferred otherwise.

Thank you for the comment. We included a sentence in the discussion to address this information (lines 268-272).

Specific comments:

1. Lines 69-70. In the context of CKD, it seems prudent to list inflammation and malnutrition as additional factors that could influence hypoalbuminemia.

Thank you for your comment. We included this information and a reference (line 70).

2. Line 80. Please abbreviate body condition score (BCS) here and then use this abbreviation subsequently (eg, line 116)

Thank you for pointing this out. We included the abbreviation (line 81).

3. Line 100. Please denote that MMS refers to muscle mass score here before its first abbreviation. See line 116.

Thank you for pointing this out. We included the abbreviation (line 81).

Reviewer #2: You have written a very nice paper here with some very interesting results. I thoroughly enjoyed reading it. One thing which does spring to mind which maybe a study worth undertaking is to define reference ranges for these biochemical parameters which will allow for comparison of studies between countries. I have only made minor points as I think the manuscript is good, and well written, so I both commend you, and thank you.

Line 34- influenced survival?

Thank you for pointing this out. We corrected the sentence (lines 34).

Line 46- perhaps in the hospitalised dog population?

Thank you. We changed the writing (line 46).

Line 49- replace ‘with’ with ‘of’

Thank you for pointing this out. We corrected the sentence (line 49).

Table 1- is this the most up to date details? I believe that this has been updated (but could be wrong)

I am also not convinced that you need this, but If you wish you can leave it in

Thank you for your comment. As also stated by the other reviewer, we have accepted the suggestion and removed the table.

Line 75- Maybe state what you class as the reference range as in my experience this seems to vary slightly between countries and even between clinics

In this specific sentence, we refer to the reference ranges cited in the references. We included some words in the sentence to make it more clear to the readers (line 75-76).

Line 89- those of ideal weight

Thank you for pointing this out. We corrected the sentence (line 91).

Line 99- you perhaps need to define BCS earlier?

Thank you for pointing this out. We corrected this and cited BCS earlier in the text (line 81).

Line 100- possibly also need to define MMS?

Thank you for pointing this out. We corrected this and cited MMS earlier in the text (line 81).

Line 110- did you have all the clinical stages of disease for each animal in the study?

We are not sure if we understand this questioning. If you mean if we followed-up on animals and recorded if they progressed in the disease, the answer is no. As this was a retrospective study, and the main point was to evaluate the animals at the time of diagnosis, information regarding staging and progression of the disease was not obtained. If you ask if we had animals in all CKD stages, we included only animals stages 2, 3 and 4.

Line 116- you have the above two terms defined here- these would be better prior to this

Thank you for pointing this out. We corrected it, as they are cited earlier in the text (line 119).

Line 117- maybe better reworded as ‘if feeding tubes were used’?

Thank you for pointing this out. We corrected the sentence (line 120).

Line 127- p = <0.05

Thank you for pointing this out. We corrected the sentence (line 136).

Line 130- looks like a space missing between ‘and 116’

Thank you for pointing this out. We corrected the sentence (line 139).

Line 132- I wonder if the mean for the body weight is the best, or would a range be better? As a miniature schnauzer would be much less than a golden retriever? Were any differences seen in biochemical parameters between breeds? Or any difference in survival?

Thank you for your comment. We included the range of age and body weight in the manuscript (lines 141-141). As for the comparison of breeds, we did not analyze this parameter because there is a great variance between the group sizes.

Line 140- at the time of data analysis?

Thank you for pointing this out. We corrected the sentence (line 149).

Table 2- did you take into account variation among people for BSC and MMS? That may be very different between animals

We obtained the BCS and MMS from the records, and evaluations of these parameters were performed by different veterinarians. Although we understand that these parameters may be subjective, we have highly trained veterinarians in our Nutrology Service with experience, and therefore we trust our records to be as close to reality as possible.

Also, how did you define hyporexic and anorexic?

We included a sentence to make it more clear as how we defined anorexic and hyporexic animals (lines 123-126).

Line 171- also influenced survival?

Thank you for pointing this out. We corrected the sentence (line 180).

Line 180- at the time of diagnosis?

Thank you for pointing this out. We corrected the sentence (line 194).

Line 180-181- Would it be possible to have more details on the contents of these foods? Perhaps as a supplementary table? If not, I think somewhere it would be nice to define what you mean by a renal diet and what that will include.

As this was a retrospective study including cases from 2013 to 2018, it is not possible to include a table with the guaranteed analysis of all the diets used since manufactures change their formulation from time to time, and this information could not be obtained. However, we included the information in the legend of Table 1 (line 160).

I do wonder if the figures would be better as a large block rather than individual? But am happy either way so will leave that up to you

Thank you for your comment. We, however, opted to keep the figures as individuals.

Line 203- no alteration was seen in appetite and …

We opted to maintain the sentence as it was. In this sentence, we are referring to “no alteration in appetite” as one of the factors that were associated with survival.

Line 221- at the time of diagnosis?

Thank you for pointing this out. We corrected the sentence (line 238).

Line 229- 235- I think this is an important study which needs to be done at some point, to define exact parameters for this disease which allows for cross study comparison, and similar treatment and diagnostics of dogs between surgeries

We agree. Studies to better define important parameters, especially in common illnesses, are necessary to improve our diagnostics and therefore treatment in veterinary medicine.

Line 243- haematocrit values of 37.0% of higher….

Thank you for pointing this out. We corrected the sentence (line 260).

Line 258- I think this section ends very abruptly. Perhaps a nice summary line may be useful?

Thank you for your comment. We added one sentence to summarize the paragraph (lines 280-282). 

References

IRIS. Sataging of CKD. Disponível em: <http://www.iris-kidney.com/pdf/IRIS_2017_Staging_of_CKD_09May18.pdf>. Access in: 2 dez. 2019. 

MICHEL, K. E. et al. Correlation of a feline muscle mass score with body composition determined by dual-energy X-ray absorptiometry. British Journal of Nutrition, v. 106, p. S57–S59, 2011. 

FREEMAN, L. M. et al. Nutritional Assessment Guidelines. Journal of Small Animal Practice, v. 00, p. 1–12, 2011.

---

## [Editor Report · Decision Letter 1]

2 Jun 2020

Nutritional and laboratory parameters affect the survival of dogs with chronic kidney disease

PONE-D-20-08705R1

Dear Dr. Brunetto

We are pleased to inform you that your manuscript has been judged scientifically suitable for publication and will be formally accepted for publication once it complies with all outstanding technical requirements.

With kind regards,

Simon Clegg, PhD

Academic Editor

PLOS ONE

Additional Editor Comments (optional):

Many thanks for resubmitting your manuscript to PLOS One and for the detailed response to reviewers.

I have reviewed the manuscript, and as all comments had been addressed, I have recommended your manuscript for publication

You should hear from the Editorial Office soon

It was a pleasure working with you, and I wish you all the best for your future research

Hope you are keeping safe and well in these difficult times

Thanks

Simon

---

## [Editor Report · Acceptance letter]

19 Jun 2020

PONE-D-20-08705R1 

Nutritional and laboratory parameters affect the survival of dogs with chronic kidney disease 

Dear Dr. Brunetto:

I'm pleased to inform you that your manuscript has been deemed suitable for publication in PLOS ONE. Congratulations! Your manuscript is now with our production department. 

Kind regards, 

on behalf of

Dr. Simon Clegg 

Academic Editor

PLOS ONE